# Tracking the progress of language models by extracting their underlying knowledge graphs

## Abstract

The state of the art of language models, previously dominated by pre-trained word embeddings, is now being pushed forward by large pre-trained contextual representations. This success has driven growing interest to understand what these models encode inside their inner workings. Despite this, understanding their semantic skills has been elusive, often leading to unsuccessful, non-conclusive, or contradictory results among different works. In this work, we define a probing classifier that we use to extract the underlying knowledge graph of nine of the currently most influential language models, including word embeddings, context encoders, and text generators. This probe is based on concept relatedness, grounded on WordNet. Our results show that this knowledge is present in all the models, but has several inaccuracies. Furthermore, we show that the different pre-training strategies and architectures lead to different model biases. We conduct a systematic evaluation to discover specific factors that explain why some concepts are challenging for the different families of models. We hope our insights will motivate the future development of models that capture concepts more precisely.

## 1 Introduction

Natural language processing (NLP) encompasses a wide variety of applications such as summarization (Kovaleva et al., 2019), information retrieval (Zhan et al., 2020), and machine translation (Tang et al., 2018), among others. Currently, the use of pre-trained language models has become the *de facto* starting point to tackle most of these applications. The usual pipeline consists of finetuning a pre-trained language model by using a discriminative learning objective to adapt the model to the requirements of each specific task. As key ingredients, these models are pre-trained using massive amounts of unlabeled data that can include millions of documents, and may include billions of parameters. Massive data and parameters are supplemented with a suitable learning architecture, resulting in a highly powerful but also complex model, whose internal operation is hard to analyze.

The success of pre-trained language models has driven the interest to understand how they manage to solve NLP tasks. As an example, in the case of BERT (Devlin et al., 2019), one of the most popular pre-trained models based on a Transformer architecture (Vaswani et al., 2017), several studies have attempted to access the knowledge encoded in its layers and attention heads (Tenney et al., 2019b; Devlin et al., 2019; Hewitt & Manning, 2019). In particular, (Jawahar et al., 2019) shows that BERT can solve tasks at a syntactic level by using Transformer blocks to encode a soft hierarchy of features at different levels of abstraction. Similarly, (Hewitt & Manning, 2019) shows that BERT is capable of encoding structural information from text. In particular, using a structural probe, they show that syntax trees are embedded in a linear transformation of the encodings provided by BERT.

In general, previous efforts have provided strong evidence indicating that current pre-trained language models encode complex syntactic rules, however, relevant evidence about their abilities to capture semantic information remains still elusive. As an example, a recent study (Si et al., 2019) attempts to locate the encoding of semantic information as part of the top layers of Transformer architectures, however, results provide contradictory evidence. Similarly, (Kovaleva et al., 2019) focuses on studying knowledge encoded by self-attention weights, however, results provide evidence for over-parameterization but not about language understanding capabilities.

In this work, we study to which extent pre-trained language models encode semantic information. As a key source of semantic knowledge, we focus on studying how precisely pre-trained language models encode the concept relations embedded in the conceptual taxonomy of WordNet[1] (Miller, 1995). The ability to understand, organize, and correctly use concepts is one of the most remarkable capabilities of human intelligence (Lake et al., 2017). Therefore, a quantification of the ability that a pre-trained language model can exhibit to encode the conceptual organization behind WordNet is highly valuable. In particular, it can provide useful insights about the inner mechanisms that these models use to encode semantic information. Furthermore, an analysis of concepts and associations that result difficult to these models can provide relevant insights about how to improve them.

In contrast to most previous works, we do not focus on a particular model, but target a large list of the most popular pre-trained language and text-embedding models. In this sense, one of our goals is to provide a comparative analysis of the capacities of different types of approaches. Following Hewitt & Manning (2019), we study semantic performance by defining a probing classifier based on concept relatedness according to WordNet. Using this tool, we analyze the different models, enlightening how and where semantic knowledge is encoded. Furthermore, we explore how these models encode suitable information to recreate the structure of WordNet. Among our main results, we show that the different pre-training strategies and architectures lead to different model biases. In particular, we show that contextualized word embeddings, such as BERT, encode high-level concepts and hierarchical relationships among them, creating a taxonomy. This finding corroborates previous work results (Reif et al., 2019) that claim that BERT vectors are stored in sub-spaces that have correspondence with semantic knowledge. Our study also shows evidence about the limitations of current pre-trained language models, demonstrating that they all have difficulties encoding specific concepts. As an example, all the models struggle with concepts related to "taxonomical groups", performing worse than chance in some cases. Our results also reveal that models have very distinctive patterns in terms of where they encode most of the semantic information. These patterns are dependant on architecture and not on model sizes.

## 2 RELATED WORK

The success of deep learning architectures in various NLP tasks has fueled a growing interest from the community to improve understanding of what these models encode. Several works have studied these models' impact on downstream tasks at the syntactic or semantic level. Some studies (Tenney et al., 2019b) claim that success in a specific task helps understand what type of information the model encodes. Other studies have improved the understanding of what and where these models encode information, by analyzing correlations between input-targets and specific architecture blocks, such as layers (Jawahar et al., 2019), encoded hidden states (Tang et al., 2018; Saphra & Lopez, 2019), and attention heads (Michel et al., 2019).

**Evidence of syntactic information**: Using probing classifiers, Clark et al. (2019) claims that some specific BERT's attention heads show correspondence with syntactic tasks. Goldberg (2019) illustrates the capabilities that BERT has to solve syntactic tasks, such as subject-verb agreement. BERT's success in these tasks fuels the belief that BERT can code the syntax of a language. Hewitt & Manning (2019) proposes a structural probe that evaluates whether syntax trees are encoded in a linear transformation of BERT embeddings. The study shows that such transformation exists in BERT, providing evidence that syntax trees are implicitly embedded in BERT's vector geometry. Reif et al. (2019) has found evidence of syntactic representation in BERT's attention matrices, with specific directions in space representing particular dependency relations.

**Evidence of semantic information**: Reif et al. (2019) suggests that BERT's internal geometry may be broken into multiple linear subspaces, with separate spaces for different syntactic and semantic information. Despite this, previous work has not yet reached consensus about this topic. While some studies show satisfactory results in tasks such as entity types (Tenney et al., 2019a), semantic roles (Rogers et al., 2020), and sentence completion (Ettinger, 2020), other studies show less favorable results in coreference (Tenney et al., 2019b) and Multiple-Choice Reading Comprehension (Si et al., 2019), claiming that BERT's performance may not reflect the model's true ability of language understanding and reasoning. Some works have studied which blocks of BERT are used to solve

---

[1] WordNet is a human-generated graph, where each one of its 117000 nodes (also called synsets) represent a concept. In this work we use the hyponymy relations, which represent if a concept is a subclass of another.

tasks at the semantic level. Tenney et al. (2019b) proposes a set of edge probing tasks to test the encoded sentential structure of contextualized word embeddings. The study shows evidence that the improvements that BERT and GPT-2 offer over non contextualized embeddings as GloVe is only significant in syntactic-level tasks. Regarding static word embeddings, Yaghoobzadeh et al. (2019) shows that senses are well represented in single-vector embeddings if they are frequent, and that this does not have a negative impact on NLP tasks whose performance depends on frequent senses.

**Layer-wise or head-wise information**: Tenney et al. (2019a) shows that the first layers of BERT focus on encoding short dependency relationships at the syntactic level (e.g., subject-verb agreement) while top layers focus on encoding long-range dependencies (e.g., subject-object dependencies). Peters et al. (2018a) supports similar declarations for Convolutional, LSTM, and self-attention architectures. While these studies also support that the top layers appear to encode semantic information, the evidence to support this claim is not conclusive or contradictory with other works. As an example, Jawahar et al. (2019) could only identify one SentEval semantic task that topped at the last layer. In terms of information flow, Voita et al. (2019a) reports that information about the past in left-to-right language models gets vanished as the information flows from bottom to top BERT's layers. Hao et al. (2019) shows that the lower layers of BERT change less during finetuning, suggesting that layers close to inputs learn more transferable language representations. In terms of architecture design, Press et al. (2020) provides evidence that increasing self-attention at the bottom and increasing feed-forward sub-layers at the top improves results in language modeling tasks using BERT. Other studies have focused on understanding how self-attention heads contribute to solving specific tasks (Vig, 2019). Kovaleva et al. (2019) shows a set of attention patterns that is repeated across different heads when trying to solve GLUE tasks (Wang et al., 2018). Furthermore, Michel et al. (2019) and Voita et al. (2019b) show that several heads can be removed from trained Transformer models without degradation in downstream tasks.

In summary, on the one hand, related work shows results that provide strong evidence concluding that BERT and other Transformer-based models can encode information at the syntactic level. Furthermore, BERT uses this information to solve various benchmark NLP tasks. Working with ambiguous words has allowed BERT to perform well on Machine Translation tasks (MT), and outperforming other architectures in word sense disambiguation (WSD). On the other hand, while some studies claim that top layers are helpful to solve semantic tasks, the results are not conclusive. Efforts to locate which blocks of the Transformer architecture operate at the semantic level have been unsuccessful. In this context, our work helps to fill the gap between the deeper understanding that we have now about how word embeddings and language models encode and work with syntax, and the still shallow comprehension of their abilities to encode and work with semantics.

## 3 STUDY METHODOLOGY

Probing methods consist of using the representation of a frozen pre-trained model to train a classifier to achieve a particular task. If the probing classifier succeeds in this setting but fails using an alternative model, it means that the source model encodes the knowledge needed to solve the task. Furthermore, the performance of the classifier can be used to measure how well the model captures this knowledge (Conneau et al., 2018). Following Hewitt & Manning (2019), we use a similar probing method at the semantic level and apply it to the nine models presented in Section 3.1. Probing techniques have shown useful for NLP tasks (Saphra & Lopez, 2019), however, they have also been questioned (Ravichander et al., 2020), as probe success does not guarantee that the model relies on that information to solve a target task. Consequently, our study limits to shed light on whether the models under evaluation encode relevant knowledge to solve the task of predicting concept relatedness in a semantic ontology such as Wordnet.

To study how precisely the models encode semantic information, we measure correctness in predicted relations among concepts at two levels: (a) pair-wise-level by studying performance across sampled pairs of related or unrelated concepts, and (b) graph-level by using pair-wise predictions to reconstruct the actual graph. We describe these two approaches in Sections 3.2 and 3.3, respectively. They essentially study the same phenomenon as both share the exact same model predictions. They only differ in the way they use these predictions.

### 3.1 WORD EMBEDDINGS MODELS

This study considers the most influential language models from recent years. We consider the essential approaches of three model families: non contextualized word embeddings (NCE), contextualized word embeddings (CE), and generative language models (GLM). We consider Word2Vec (Mikolov et al., 2013) and GloVe (Pennington et al., 2014) for the first family of approaches. For the CE family, we consider ELMo (Peters et al., 2018b), which is implemented on a bidirectional LSTM architecture, XLNet (Yang et al., 2019), and BERT (Devlin et al., 2019) and its extensions ALBERT (Lan et al., 2020) and RoBERTa (Liu et al., 2019), all of them based on the Transformer architecture. GPT-2 (Radford et al., 2018) and T5 (Raffel et al., 2019) are included in the study to incorporate approaches based on generative language models.

### 3.2 SEMANTIC PROBING CLASSIFIER

We define an edge probing classifier that learns to identify if two concepts are semantically related. To create the probing classifier, we retrieve all the glosses from the Princeton WordNet Gloss Corpus[2]. This dataset provides WordNet's synsets gloss sentences with annotations identifying occurrences of concepts within different sentence contexts. The annotations provide a mapping of the used words to their corresponding WordNet node. We sample hypernym pairs A, B. Then, from an unrelated section of the taxonomy, we randomly sample a third synset C, taking care that C is not related to either A or B. Then, $\langle A, B, C \rangle$ forms a triplet that allows us to create six testing edges for our classifier. To train the probing classifier, we define a labeled edge $\{x, y, L\}$, with $x$ and $y$ synsets in $\{A, B, C\}$, $x \neq y$. $L \in \{0, 1\}$ is the target of the edge. If $y$ is direct or indirect parent of $x$, $L = 1$, while $L = 0$ in other case. For each synset $x, y$, we sample one of its sentences $S(x)$, $S(y)$ from the dataset. Let $M$ be a word embedding model. If $M$ belongs to the NCE family, $x$ and $y$ are encoded by $M(x)$ and $M(y)$, respectively. If $M$ belongs to the CE or GLM families, then $x$ and $y$ are encoded by the corresponding token of $M(S(x))$ and $M(S(y))$, respectively. Accordingly, the context of each concept provides additional information for the creation of its word embedding.

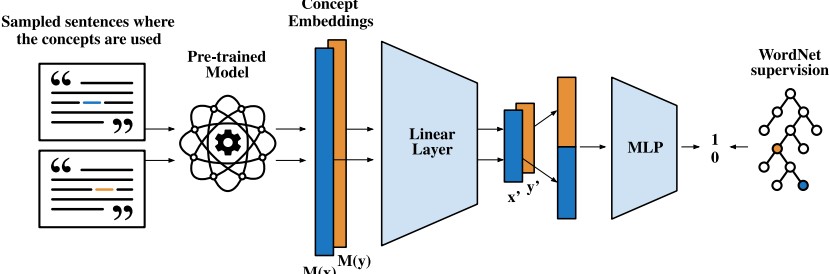

Figure 1: Inputs to the edge probing classifier correspond to the model embeddings $M(x)$ and $M(y)$ of concepts x and y, respectively. To account for the different dimensionality of the tested models, $M(x)$ and $M(y)$ are projected into a common lower dimensionality space using a linear layer. The resulting embeddings $x'$ and $y'$ are concatenated into a vector, which is fed into a Multi-Layer Perceptron that is in charge of predicting if the concept pair is related or not.

To accommodate for the evaluation of concept embeddings of different dimensionalities, we first project the embeddings of each concept $x$ and $y$ into a low dimensionality space using a linear layer (see Figure 1). Choosing a low target dimensionality for this linear layer was also essential to avoid overfitting the classification module. Then, these vectors, denoted as $x'$ and $y'$, are concatenated and fed into a Multi-Layer Perceptron (MLP) classifier. The linear layer and the MLP are the only trainable parameters of our setting, as we use the source model weights without any finetuning.

Similar structural probes have been proposed to evaluate syntax encoding in BERT. However, we use a MLP to learn the structural relation between concept pairs, providing the test with a mechanism that allows the embeddings to be combined in a non-linear way. Tests based on linear transformations such as the one proposed by Hewitt & Manning (2019) did not allow us to recover the WordNet

---

[2]https://wordnetcode.princeton.edu/glosstag.shtml

structure. This indicates that the sub-spaces where the language models encode this semantic information are not linear. The fact that syntactic information is linearly available with high accuracy, makes us believe that syntax trees might be a critical intermediate result needed for the language modeling task; whereas semantic information emerges as an indirect consequence of accurate language modeling and syntactic identification, but might not constitute information that the model relies on for the language modeling task, as postulated by Ravichander et al. (2020).

### 3.3 RECONSTRUCTING THE STRUCTURE OF A KNOWLEDGE GRAPH

The proposed probe classifier predicts if a pair of concepts $\langle u, v \rangle$ form a valid $\langle \text{parent}, \text{child} \rangle$ relation according to WordNet, where $h_{\langle u,v \rangle} \in [0, 1]$ denotes the corresponding classifier output. It is important to note that valid $\langle \text{parent}, \text{child} \rangle$ relations include direct relations, as in $\langle \text{dog}, \text{poodle} \rangle$, and transitive relations, as in $\langle \text{animal}, \text{poodle} \rangle$, and that the order of the items in the relation do matter.

To reconstruct the underlying knowledge graph, for each valid $\langle \text{parent}, \text{child} \rangle$ relation given by $h_{\langle u,v \rangle} > threshold$, we need an estimation of how close are the corresponding nodes in the graph. To do this, we introduce the concept of "parent closeness" between a parent node $u$ and a child node $v$, that we define as $d_e(u, v)$. We propose two alternative scores to estimate $d_e$:

**i) Transitive Intersections Metric (TIM)**: we explore a metric grounded directly in the tree structure of a knowledge graph. The key observations are that nodes $u$ and $v$ that form a parent-child relation have some transitive connections in common. Specifically, all descendants of $v$ are also descendants of $u$, and all the ancestors of $u$ are also ancestors of $v$. Furthermore, the closer the link between $u$ and $v$ in the graph, the bigger the intersection. With that in mind, for each candidate edge $e = \langle u, v \rangle$ we define the following metric for $d_e$:

$$d_e(u, v) = C - \left( \sum_{j \in N \setminus \{u,v\}} h_{\langle u,j \rangle} h_{\langle v,j \rangle} + \sum_{j \in N \setminus \{u,v\}} h_{\langle j,u \rangle} h_{\langle j,v \rangle} \right) * h_{\langle u,v \rangle} \tag{1}$$

where the first sum accounts for the similarity in the descendants of nodes $u$ and $v$, and the second sum accounts for the similarity in the ancestors of nodes $u$ and $v$. The term $h_{\langle u,v \rangle}$ at the right-hand side accounts for the direction of the edge. Constant $C = \max_{u_i,v_j} d(u_i, v_j)$ is added to force $d_e$ to be positive. $N$ denotes the set of all nodes (concepts).

**ii) Model Confidence Metric (MCM)**: All nine models in this study capture close concept relations more precisely than distant relations (supporting evidence can be found in Appendix C). In practice, this means that a concept like *poodle* will be matched with its direct parent node *dog* with higher confidence than with a more distant parent node such as *animal*. Thus it is reasonable to use $d_e(u, v) = 1 - h_{\langle u,v \rangle}$.

Based on $d_e$, a natural choice to find the tree that comprises only the closest parent of each node is to obtain the minimum-spanning-arborescence (MSA) of the graph defined by $d_e$. The MSA is analogous to the minimum-spanning-tree (MST) objective used in Hewitt & Manning (2019), but for the specific case of directed graphs. For a mathematical formulation of the MSA optimization problem applied to our setting, we refer readers to Appendix A.3.

## 4 HOW ACCURATE IS THE SEMANTIC KNOWLEDGE?

### 4.1 SEMANTIC EDGE PROBING CLASSIFIER RESULTS

Table 1 shows the results obtained using the edge probing classifier in several model variants. Results show that regardless of model sizes, performance is homogeneous within each family of models. Additionally, results show that when all the layers are used, NCE and GLM methods obtain a worse performance than those achieved by CE methods. When single layers are used, GLM show improved performance, suggesting that the generative objective forces the model to capture semantics earlier in the architecture in order to keep the last layers for generative-specific purposes that do not have a direct relationship with this kind of knowledge. In contrast, CE models degrade or maintain their performance when single layers are used.

Notice that results in Table 1 show pair-wise metrics not graph metrics. As we are dealing with graphs, predicted edges are built upon related predicted edges, thus drifts in small regions of the

| Family | Model | Emb. Size All/Best Layer | Best Layer | F1-score All Layers | F1-score Best Layer |
|---|---|---|---|---|---|
| NCE | Word2Vec (Mikolov et al., 2013) | 300 / - | - | .7683 ± .0135 | - |
| | GloVe-42B (Pennington et al., 2014) | 300 / - | - | .7877 ± .0084 | - |
| GLM | GPT-2 (Radford et al., 2018) | 9984 / 768 | 6 | .7862 ± .0132 | .7921 ± .0108 |
| | T5-small (Raffel et al., 2019) | 7168 / 512 | 4 | .8156 ± .0098 | .8199 ± .0081 |
| | GPT2-xl (Radford et al., 2018) | 78400 / 1600 | 13 | .7946 ± .0151 | .8029 ± .0118 |
| | T5-large (Raffel et al., 2019) | 51200 / 1024 | 17 | .8148 ± .0119 | .8331 ± .0102 |
| CE | ELMo-small (Peters et al., 2018b) | 768 / 256 | 2 | .7986 ± .0126 | .7880 ± .0119 |
| | BERT-base (Devlin et al., 2019) | 9984 / 768 | 10 | .8240 ± .0123 | .8185 ± .0104 |
| | RoBERTa-base (Liu et al., 2019) | 9984 / 768 | 5 | .8392 ± .0100 | .8266 ± .0083 |
| | XLNet-base (Yang et al., 2019) | 9984 / 768 | 4 | .8306 ± .0113 | .8293 ± .0116 |
| | ALBERT-base (Lan et al., 2020) | 9984 / 768 | 12 | .8184 ± .0222 | .8073 ± .0102 |
| | ELMo-large (Peters et al., 2018b) | 3072 / 1024 | 2 | .8311 ± .0090 | .8330 ± .0083 |
| | BERT-large (Devlin et al., 2019) | 25600 / 1024 | 14 | .8178 ± .0152 | .8185 ± .0113 |
| | RoBERTa-large (Liu et al., 2019) | 25600 / 1024 | 13 | .8219 ± .0159 | .8314 ± .0082 |
| | XLNet-large (Yang et al., 2019) | 25600 / 1024 | 6 | .8211 ± .0142 | .8244 ± .0080 |
| | ALBERT-xxlarge (Lan et al., 2020) | 53248 / 4096 | 4 | .8233 ± .0107 | .8194 ± .0097 |

Table 1: Results obtained using the edge probing classifier. We study the performance in many model's variants, considering small and large versions of several models. Results are shown grouped by the families of methods defined in Section 3.1.

graph may cause large drifts in downstream connections. Furthermore, our setup balances positive (links) and negative samples (not links), but the proportion of negative samples will be considerably larger in a real reconstruction scenario. As a consequence, we emphasize that these numbers must not be considered in isolation, but together with the results reported in sections 4.2 and 5.

## 4.2 EXTRACTING THE UNDERLYING KNOWLEDGE GRAPH

Predicting a knowledge graph has a complexity of at least $O(N^2)$ in the number of analyzed concepts. In our case, this imposes a highly demanding computational obstacle because WordNet has over 82000 noun synsets. To accelerate experimentation and to facilitate our analysis and visualizations, we focus on extracting a WordNet sub-graph that comprises only 46 nodes. These nodes are picked to include easily recognizable relations. We use the tree-edit-distance to evaluate how close are the reconstructed graphs to the target graph extracted from WordNet. Table 2 shows our results.

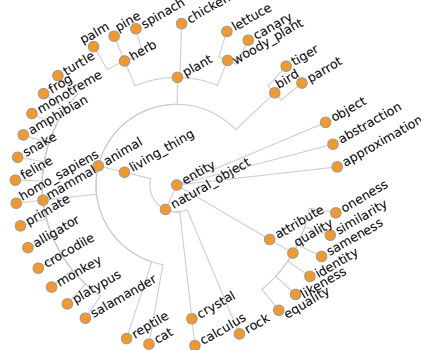

Figure 2: A reconstructed graph using MCM over BERT-large. The graphs for all the models can be found in Appendix B.

| Family | Model | Tree Edit Dist. | | |
|---|---|---|---|---|
| | | TIM | MCM | Avg. |
| NCE | Word2Vec | 59 | 59 | **59** |
| | GloVe-42B | 56 | 60 | **58** |
| GLM | GPT-2 | 53 | 57 | **55** |
| | T5 | 58 | 55 | **56** |
| CE | ELMo | 52 | 55 | **53** |
| | BERT | 49 | 48 | **49** |
| | RoBERTa | 56 | 54 | **55** |
| | XLNet | 52 | 48 | **50** |
| | ALBERT | 53 | 50 | **51** |

Table 2: Tree Edition Distance against the ground truth graph (large models used). We display both strategies for estimating $d_e$ along with their average score.

Table 2 shows that the knowledge graphs retrieved using CE models are closer to the target than graphs provided by NCE and GLM models. In particular, the best results are achieved by BERT, ALBERT, and XLNet, indicating that these models encode more accurate semantic information than the alternative models. These results are consistent with those obtained in Section 4.1.

Visual inspection of the reconstructed graphs as the one found in Figure 2 reveals that models capture some key categories but fail to map some fine-grained relations.

# 5 WHAT IS EASY OR HARD? WHAT ARE THESE MODELS LEARNING?

Section 4 shows that different model families differ in their errors. Furthermore, it shows that within the same family, models have similar biases. In this section, we elucidate which semantic factors have an impact on the performance of these models and which ones do not affect their F1-score.

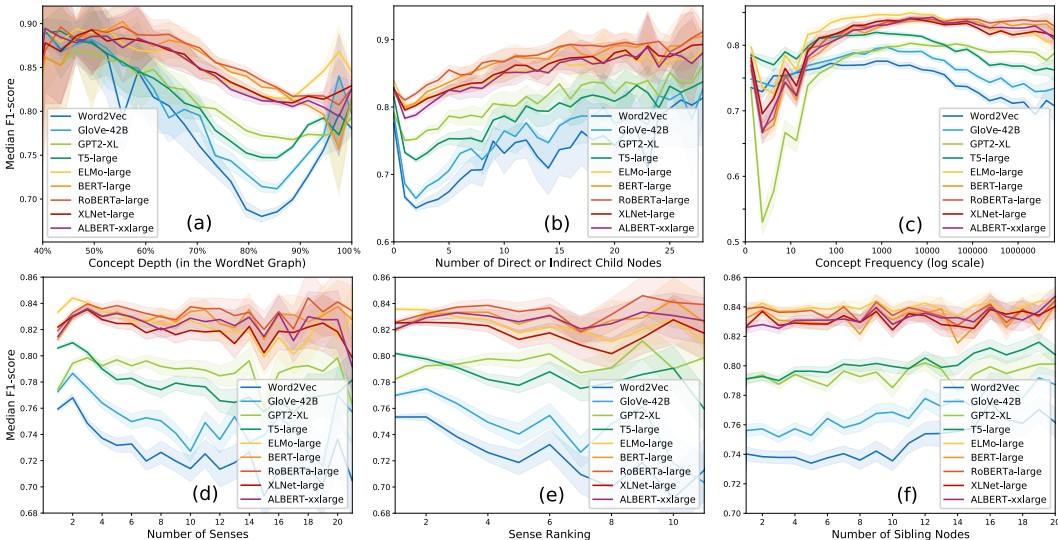

Figure 3: Semantic factors with a high (top charts) or low (bottom charts) impact on F1-score, along with their 90% confidence intervals. Charts only display ranges where at least 100 samples existed. Appendix C shows additional factors along with the specific implementation details.

Figure 3-a shows that most models decrease their F1-score as concepts get more specific (increased depth). We hypothesize that this is because higher-level concepts (e.g. *Animal*) appear more frequently and in more diverse contexts, as they are also seen as instances of their subclasses (e.g. *Dog*, *Cat*, *Chihuahua*), allowing the models to learn more precise representations for them. In contrast, lower-level concepts will only be present in their specific contexts (e.g. texts about *Apple-Head-Chihuahua*). This hypothesis is confirmed by Figure 3-b, as concepts with higher number of sub-classes have higher F1-scores. Furthermore, Figure 3-c also shows that models tend to capture concepts more accurately as the model is exposed to them more frequently in the pre-training corpus.

Additionally, Figure 3-c indicates that the models degrade their F1-score when concepts are too frequent. In particular, NCE and GLM models are more sensitive to this factor. We hypothesize that this is because high-frequency concepts appear in contexts with too much variability. Their unspecific use makes more difficult to distinguish accurate meaning for NCE and GLM models.

Another surprising finding is that CE and GLM models are almost unaffected by the number of senses that a certain word has, neither to their sense ranking or their number of sibling concepts, displaying almost flat charts (Figure 3-d-e-f). This finding suggests that these models pay as much attention to the context as to the target word, and are probably biased in favor of contextual information. This is intuitive for Masked-Language-Models such as BERT, but not for others, such as GPT-2. Furthermore, this behavior is opposed to what NCE models exhibit according to Yaghoobzadeh et al. (2019) and our experiments, as NCE models tend to focus more on the most frequent senses.

Analysis of specific semantic categories indicates that in most cases, the models of the same family have similar behaviors, especially within the NCE or CE families. More importantly, different families show different patterns. Table 3 shows some salient examples. Surprisingly, all models struggle with the category "taxonomic groups". Manual inspection of sentences and analysis of related concepts make us believe that in these cases the use of the context confuses CE and GLM models, as in

| Family | Model | artifact | attribute | living thing | matter | person | social group | taxonomic group |
|---|---|---|---|---|---|---|---|---|
| NCE | Word2Vec | .7120 | .7044 | .7295 | .7402 | .7208 | .7497 | .6920 |
| | GloVe-42B | .7389 | .7213 | .7421 | .7633 | .7351 | .7579 | .6648 |
| GLM | GPT-2 | .7903 | .7730 | .7300 | .7582 | .7207 | .8155 | .3030 |
| | T5 | .7868 | .7649 | .7862 | .8002 | .7735 | .7868 | .6944 |
| CE | ELMo | .8308 | .8093 | .8187 | .7756 | .8022 | .8312 | .6011 |
| | BERT | .8249 | .8094 | .7593 | .7645 | .7379 | .8516 | .4804 |
| | RoBERTa | .8315 | .8167 | .7823 | .7614 | .7585 | .8552 | .4921 |
| | XLNet | .8319 | .8064 | .7907 | .7636 | .7779 | .8422 | .5371 |
| | ALBERT | .8231 | .8050 | .7758 | .7685 | .7826 | .8556 | .4277 |

Table 3: Average F1-score for some semantic categories revealing models strengths and weaknesses. Several other categories are reported in Appendix G along with their standard deviations.

many sentences the corresponding concept could be nicely replaced by another, conveying a modified but still valid message. This phenomenon does not take place in other categories such as "social group" or "attribute", even though these concepts are closely related to "taxonomic groups".

# 6 WHERE IS THIS KNOWLEDGE LOCATED?

As mentioned in Section 2, prior work has not shown consensus about where is semantic information primarily encoded inside these architectures. Our experiments shed light on this subject. Figure 4 shows how each layer contributes to the F1-score.

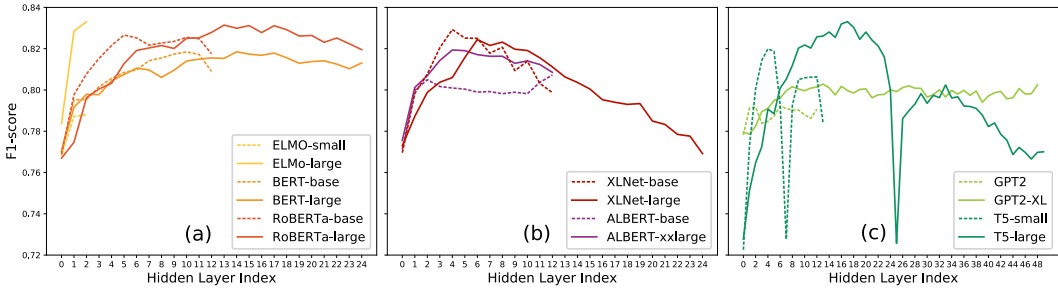

Figure 4: F1-score for hypernym prediction across each model layer.

Figures 4-a and 4-b show the performance across layers for the CE-based models. They reveal that while BERT and RoBERTa use their top-layers to encode semantic information, XLNet and AL-BERT use the first layers. Figure 4-c shows that GLM models have very distinct patterns. While GPT-2 encodes semantic information in all the layers, T5 shows an M shape because its configuration considers an encoder-decoder architecture. The chart shows that T5 uses its encoder to hold most of the semantic information. We also note that small models show similar patterns as their larger counterparts, suggesting that these patterns depend on the architecture and not on model size.

# 7 FURTHER DISCUSSION AND IMPLICATIONS

Table 4 summarizes our main findings. Findings (1), (2) and (3) indicate that, to a different extent, all models encode relevant knowledge about the hierarchical semantic relations included in WordNet. However, some of this knowledge is incorrect. In particular, as we mention in Section 5, we observe that the ability to learn about a concept depends on its frequency of appearances in the training corpus and the level of specificity of its meaning. Furthermore, some concept categories result difficult for all model families, while some categories are particularly difficult for contextual models such as CE. We hypothesize that stronger inductive biases are required to capture low-frequency concepts. Furthermore, we believe that new learning approaches are needed to discriminate accurate meaning for high-frequency concepts. As expected, our findings indicate that model families have different biases leading to different behaviors. Thus, our results can illuminate further research to improve semantic capabilities by combining the strengths of each family of models. As an example,

| | Finding | Supporting Evidence | Involved Models |
|---|---|---|---|
| (1) | All models encode a relevant amount of knowledge about semantic relations in WordNet, but this knowledge contains imprecisions. | All | All |
| (2) | The ability to learn concept relations depends on how frequent and specific are the concepts, affecting some model families heavier. | Fig. 3a-c | NCE and GLM |
| (3) | Concept difficulty is usually homogeneous within each model family. Some semantic categories challenge all models. | Table 3 | All |
| (4) | Some models encode stronger semantic knowledge than others, usually according to their family. | Tables 1, 2, 3 | ELMo, BERT, RoBERTa, ALBERT, XLNet, T5 |
| (5) | Some models focus their encoding of semantic knowledge in specific layers, and not distributed across all layers. | Table 1, Fig. 4 | GLM |
| (6) | Models have distinctive patterns as to where they encode semantic knowledge. Patterns are model-specific and not size-specific. | Table 1, Fig. 4 | All |
| (7) | Model size has an impact in the quality of the captured semantic knowledge, as seen in our layer-level probe tests. | Table 1, Fig. 4 | ELMo, RoBERTa, ALBERT, GPT-2, T5 |
| (8) | Semantic knowledge does not depend on pre-training corpus size. | Tables 1, E-6 | - |
| (9) | Contextual models are unaffected by multi-sense words. | Fig. 3d-f | CE and GLM |

Table 4: Summary of our main findings and their corresponding supporting evidence.

one could combine them as ensembles or by incorporating more than a single model head, each one equipped with a different loss function (i.e., one generative approach resembling GLM-based methods and another discriminative resembling CE-based methods).

Findings (4), (5) and (6) suggest that instead of a standard finetuning of all layers of BERT according to a given downstream task, to improve semantic capabilities one could perform a task profiling to decide the best architecture for the task and also how to take advantage of it. By using only a limited number of layers or choosing a different learning rate for each layer, one could exploit the semantic knowledge that the pre-trained model carries, avoiding the degradation of this information present at the top layers, especially when using T5, XLNet, or ALBERT-large. Accordingly, recent work on adaptive strategies to output predictions using a limited number of layers (Xin et al., 2020; Liu et al., 2020; Hou et al., 2020; Schwartz et al., 2020; Fan et al., 2020; Bapna et al., 2020) would benefit from using architectures that encode knowledge in the first layers. To the best of our knowledge, these works have only used BERT and RoBERTa, achieving a good trade-off between accuracy and efficiency. Only Zhou et al. (2020) has explored ALBERT, reporting improved accuracy by stopping earlier. Our findings explain this behavior and suggest that T5 or XLNet may boot their results even further as these architectures have sharper and higher information peaks in their first layers.

Findings (7) and (8) suggest that recent success in semantic NLP tasks might be due more to the use of larger models than large corpora for pretraining. This also suggests that to improve model performance in semantic tasks, one could train larger models even without increasing the corpus size. A similar claim has been proposed by Li et al. (2020) leading to empirical performance improvements.

Finally, finding (9) is important because it suggests that contextual models pay as much attention to the context as to the target word, and are probably biased in favor of contextual information, even if they are not based on the Masked-Language-Model setting. We believe that this inductive bias could be exploited even further in the design of the underlying architecture, thus this finding might elucidate a design direction to encourage more effective learning of semantic knowledge.

# 8 CONCLUSIONS

In this work, we exploit the semantic conceptual taxonomy behind WordNet to test the abilities of current families of pre-trained language models to learn semantic knowledge from massive sources of unlabeled data. Our main conclusion is that, indeed, to a significant extent, these models learn relevant knowledge about the organization of concepts in WordNet, but also contain several imprecisions. We also notice that different families of models present dissimilar behavior, suggesting the encoding of different biases. As part of future work, we hope our study helps to inspire new ideas to improve the semantic learning abilities of current pre-trained language models.

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

# A   IMPLEMENTATION DETAILS

## A.1   EDGE PROBING CLASSIFIER DETAILS

To study the extent to which these Language Models deal with semantic knowledge, we extend the methodology introduced by Tenney et al. (2019b). In that study, the authors defined a probing classifier at the sentence level, training a supervised classifier with a task-specific label. The probing classifier's motivation consists of verifying when the sentence's encoding help to solve a specific task, quantifying these results for different word embeddings models. We cast this methodology to deal with semantic knowledge extracted from WordNet. Rather than working at the sentence level, we define an edge probing classifier that learns to identify if two concepts are semantically related.

To create the probing classifier, we retrieve all the glosses from the Princeton WordNet Gloss Corpus[3]. The dataset provides WordNet's synsets gloss with manually matched words identifying the context-appropriate sense (see Figure 5). In WordNet, each sense is coded as one of the synsets related to the concept (e.g., sense *tendency.n.03* for the word tendency). Using a synset A and its specific sense provided by the tagged gloss, we retrieve from WordNet one of its direct or indirect hypernyms, denoted as B (see Figure 6). If WordNet defines two or more hypernyms for A, we choose one of them at random. We sample a third synset C, at random from an unrelated section of the taxonomy, taking care that C is not related to either A or B (e.g., *animal.n.01*). Then, $\langle A, B, C \rangle$ form a triplet that allows us to create six testing edges for our classifier: $\langle A, B \rangle$, which is compounded by a pair of related words through the semantic relation *hypernym of*, and five pairs of unrelated words ($\langle A, C \rangle$, $\langle B, C \rangle$, $\langle B, A \rangle$, $\langle C, A \rangle$, $\langle C, B \rangle$). We associate a label to each of these pairs that show whether the pair is related or not (see Figure 6). Note that we define directed edges, meaning that the pair $\langle A, B \rangle$ is related, but $\langle B, A \rangle$ is unrelated to the relationship *hypernym of*. Accordingly, the edge probing classifier will need to identify the pair's components and the order in which the concepts were declared in the pair.

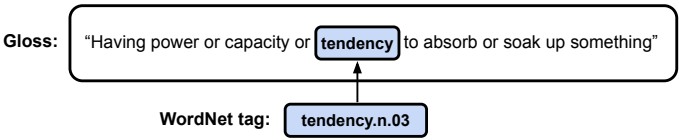

Figure 5: The Princeton WordNet Gloss Corpus provides sentences with manually annotated mappings of words to their corresponding WordNet Synset (concept / sense).

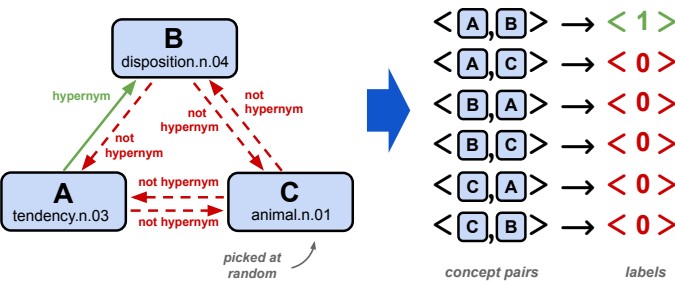

Figure 6: Each triplet is used to create related and unrelated pairs of words according to the relationship *hypernym of*. We create six edge probing pairs, and therefore, the edge probing classifier will need to identify the pair's components and the order in which the words were declared in the pair.

We create training and testing partitions ensuring that each partition has the same proportion of leaves versus internal nodes. The latter is essential to identify related pairs. During training, we guarantee that each training synset is seen at least once by the probing classifier. To guarantee the above, we sample each synset in the training set and sample some of its hypernyms at random. Then. we randomly sample some unrelated synset for each related pair that has no relation to any

---

[3]https://wordnetcode.princeton.edu/glosstag.shtml

of the words in the related pair. We create three partitions from this data on 70/15/15 for training, development, and testing foldings, respectively.

We train the MLP classifier using a weighted binary cross-entropy loss function. Since we have one positive and five negative examples per triplet, we use a weighted loss function with weights 5 and 1 for the positive and negative class, respectively. Accordingly, positive and negative examples have the same relevance during training. We implemented the linear layer and the MLP classifier using a feed forward network with 384 hidden units. The MLP was trained using dropout at 0.425 and a $L_2$ regularizer to avoid overfitting.

To create the vector representations for each of the word embeddings models considered in this study, we concatenate the hidden state vectors of all the layers for each tagged synset. For both CE and GLM-based models, each gloss was used as a context to build specific contextual word embeddings. If the gloss has more than one tagged token, we take only the first of them for the analysis.

## A.2 WORDNET METRICS: DISTANCE

Formally, let $d_W(x, y)$ be the Wordnet distance between two synsets $x$, $y$, defined by:

$$d_W(x, y) = \begin{cases} d_{\text{path}}(x, y) & \text{if } y \text{ is ancestor of } x, \\ d_{\text{path}}(x, z) + d_{\text{path}}(y, z) & \text{otherwise,} \end{cases}$$

where $d_{\text{path}}(x, y)$ is the length of the shortest path between $x$ and $y$ in WordNet, measured in number of hops, and $z$ is the closest common ancestor of $x$ and $y$ in the case that $y$ is not an ancestor of $x$.

## A.3 MINIMUM-SPANNING-ARBORESCENCE OPTIMIZATION PROBLEM APPLIED TO KNOWLEDGE GRAPH RECONSTRUCTION

Given a graph $G$ with nodes $N$ and unknown edges $E$, we define an auxiliary graph $G'$ with nodes $N$ and edges $E'$, comprised of all possible directed edges. For each edge $e \in E'$, we obtain a prediction $h_e$ that estimates the probability of that edge representing a valid hypernymy relation, and a distance $d_e$ that estimates the "parent closeness"[4] between the nodes in $G$.

We define $\delta(v)$ to be the set of edges $\{\langle u, v \rangle : u \in N, u \neq v\}$ where edge $\langle u, v \rangle$ represents a $\langle \text{parent}, \text{child} \rangle$ relation. We also define $\gamma(S)$ to be the set of edges $\{\langle u, v \rangle \in E' : u \notin S, v \in S\}$. We estimate the graph topology of $G$ defined by $E \subset E'$ by solving the following optimization problem:

$$\max_{r \in N} \sum_{e \in E'} x_e h_e \tag{2}$$

$$\text{s.t.} \quad x_e \in \arg\min \left[ \sum_{e \in E'} x_e d_e \quad \text{s.t.} \begin{cases} x_e \in \{0, 1\} & e \in E' \\ \sum_{e \in \delta(v)} x_e = 1 & \forall v \in N \setminus \{r\} \\ \sum_{e \in \gamma(S)} x_e \geq 1 & \forall S \subset N \setminus \{r\} \end{cases} \right] \tag{3}$$

Objective function (2) is used to find the best root node $r$; and the nested optimization problem (3) is the minimum spanning arborescence problem applied to the dense graph $G'$. The final binary values of $x_e$ estimate $E$ by indicating if every possible edge $e$ exist in the graph or not. To solve this optimization problem, we need estimates of $h_e$ and $d_e$ for each edge $e$. We use the output of the probing classifier as an estimate of the probability of $h_e$, and use TIM and MCM scores as estimates for $d_e$ (See Section 3.3).

---

[4]The value of this distance will be small if the hypernym relation is close, or large if it is distant or not valid.

# B   RECONSTRUCTED GRAPHS

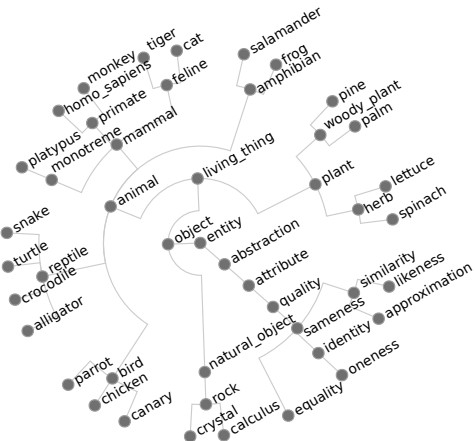

Figure 7: Ground Truth Knowledge Graph

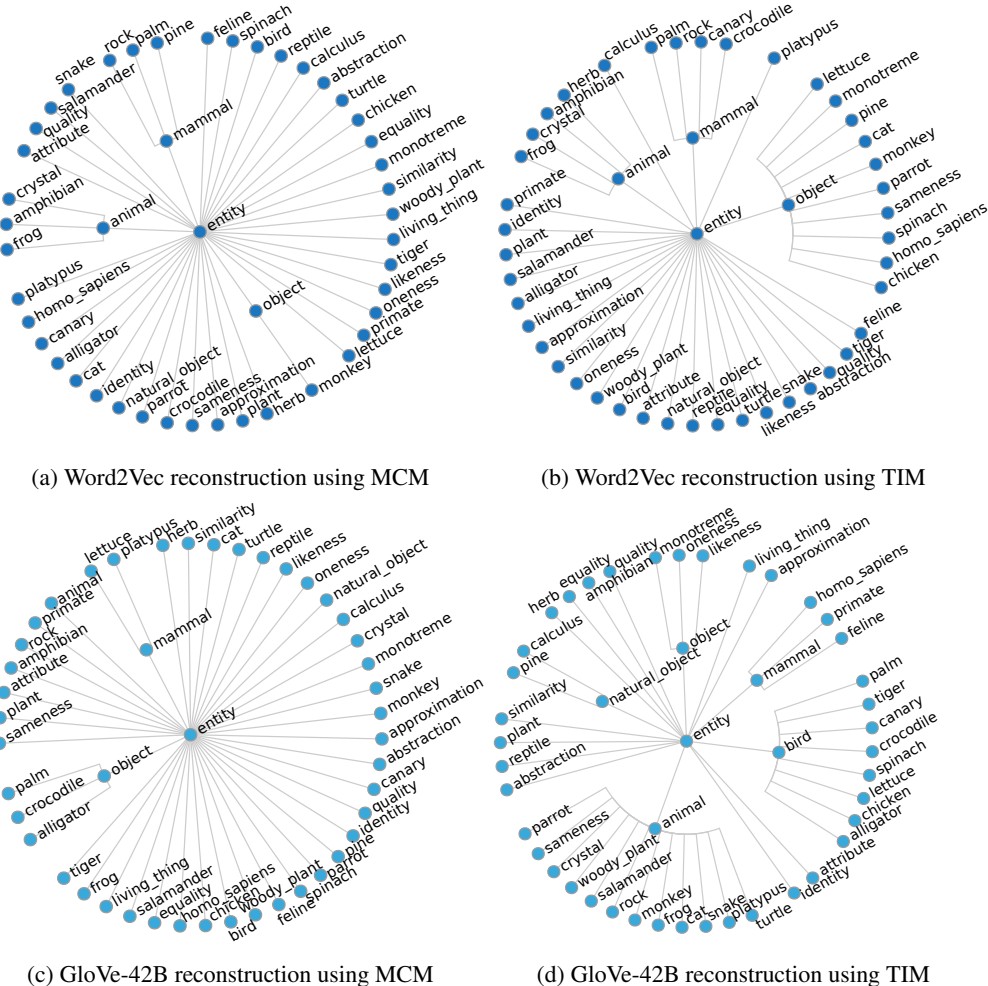

(a) Word2Vec reconstruction using MCM

(b) Word2Vec reconstruction using TIM

(c) GloVe-42B reconstruction using MCM

(d) GloVe-42B reconstruction using TIM

Figure 8: Knowledge graph reconstruction using Word2Vec and GloVe.

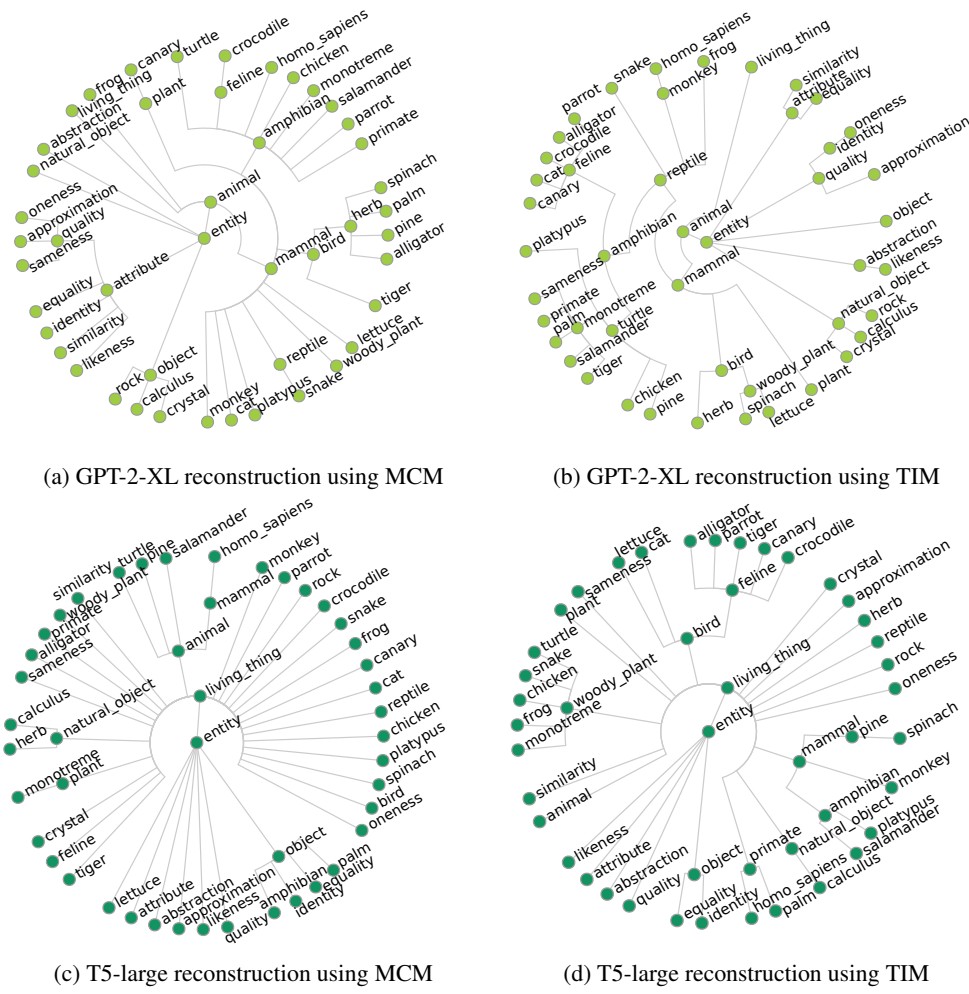

(a) GPT-2-XL reconstruction using MCM

(b) GPT-2-XL reconstruction using TIM

(c) T5-large reconstruction using MCM

(d) T5-large reconstruction using TIM

Figure 9: Knowledge graph reconstruction using GPT-2 and T5.

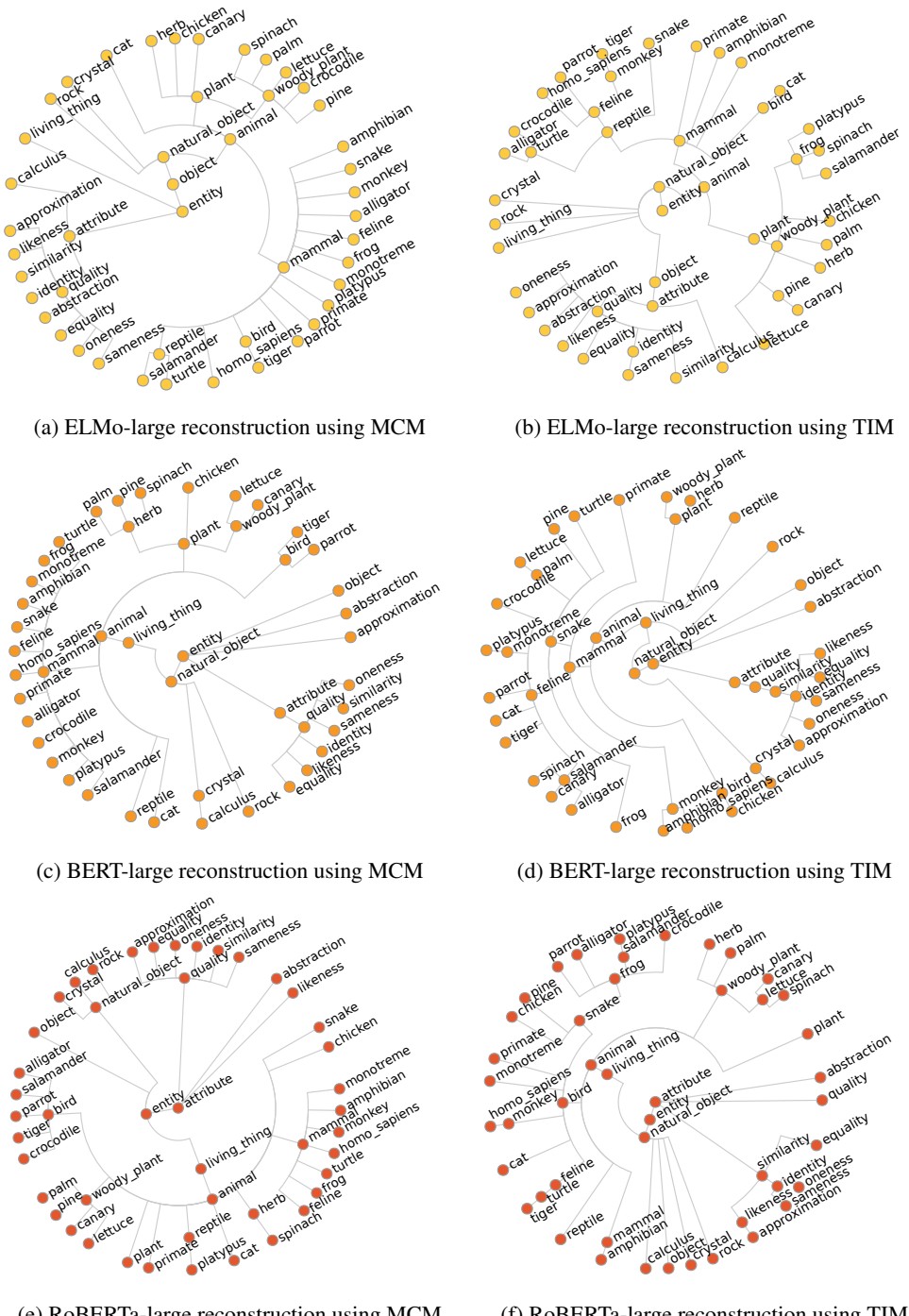

(a) ELMo-large reconstruction using MCM

(b) ELMo-large reconstruction using TIM

(c) BERT-large reconstruction using MCM

(d) BERT-large reconstruction using TIM

(e) RoBERTa-large reconstruction using MCM

(f) RoBERTa-large reconstruction using TIM

Figure 10: Knowledge graph reconstruction using ELMo, BERT and RoBERTa.

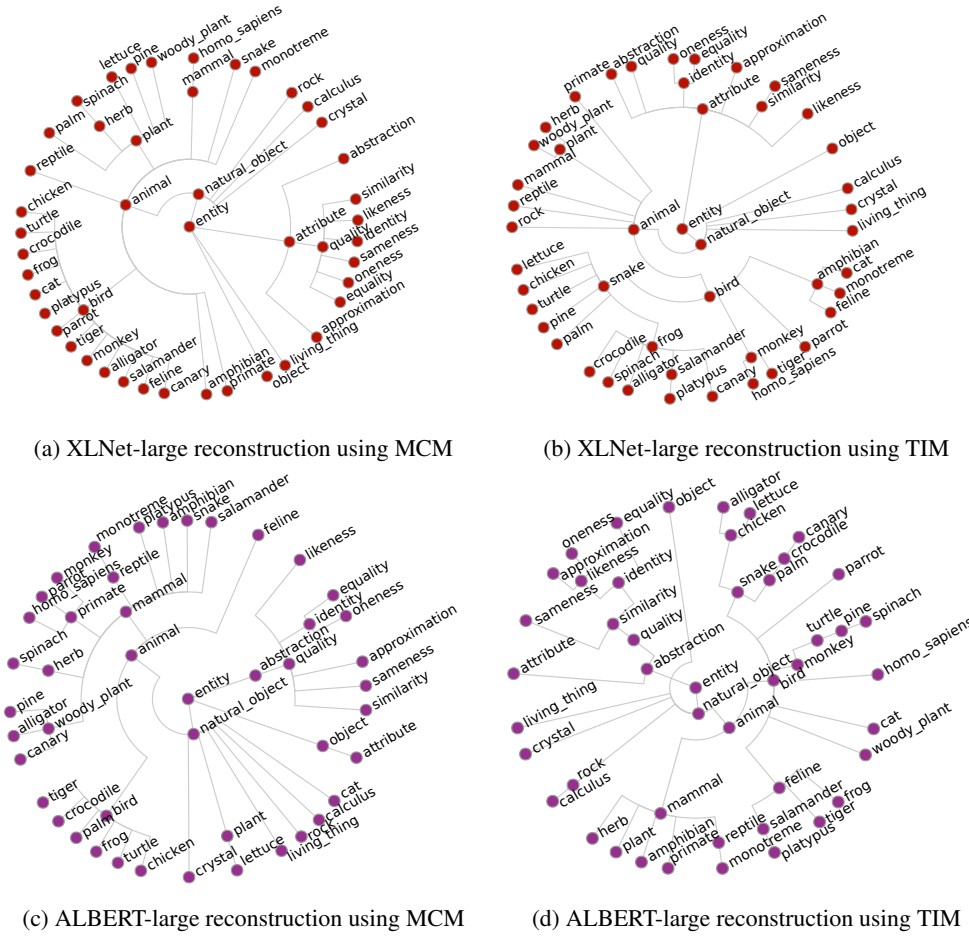

(a) XLNet-large reconstruction using MCM

(b) XLNet-large reconstruction using TIM

(c) ALBERT-large reconstruction using MCM

(d) ALBERT-large reconstruction using TIM

Figure 11: Knowledge graph reconstruction using XLNet and ALBERT.

# C  FURTHER ANALYSIS OF FACTORS' IMPACT IN MODELS PERFORMANCE

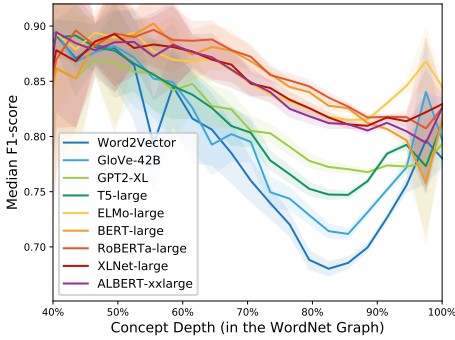

Figure 12: **Relative depth in the WordNet graph**: For each synset, we compared F1 with depth score (0 % for the root and 100 % for leaves) measuring differences between higher/lower level concepts. As seen here, NCE-based methods are significantly impacted by this variable.

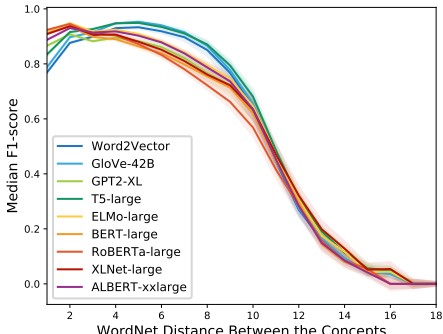

Figure 13: **Graph distance between concepts**: We measured the impact of the number of "hops" that separate two tested concepts on pair-wise F1 score. This chart reveals a strong correlation of all the models in this aspect. As an example of this phenomenon, closer relations such as ⟨chihuahua, dog⟩ are, in general, considerably easier to capture than distant relations such as ⟨chihuahua, entity⟩. For details on how we implement the distance in WordNet, check Appendix A.2.

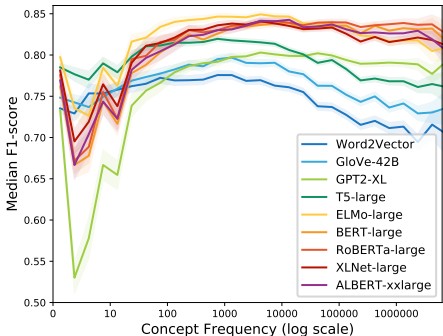

Figure 14: **Concept frequency**: In this Figure we evaluate if frequent concepts are easier or harder to capture for these models. The frequency was computed by counting occurrences in the 38 GB of OpenWebText Corpus (`http://Skylion007.github.io/OpenWebTextCorpus`). The chart shows that uncommon and highly frequent concepts are harder to be modeled. In particular, NCE-based models and T5 are more sensitive to this factor.

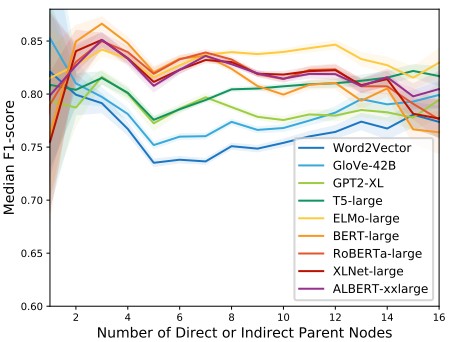
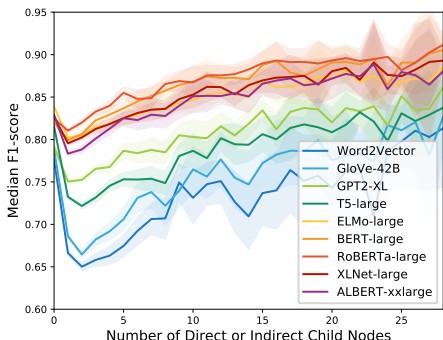

Figure 15: **Number of parent and child nodes**: We studied if the number of parents or child nodes have an impact on F1-scores. The same phenomenons show up when we analyze only direct parents or children.

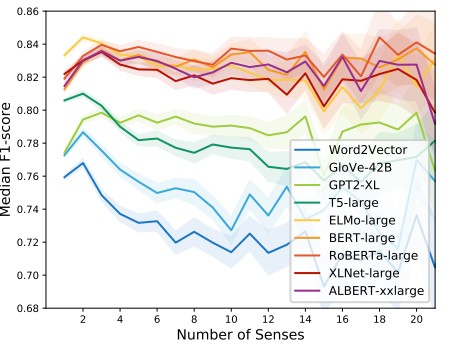
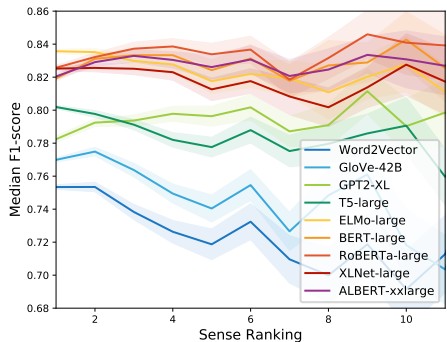

Figure 16: **Number of Senses and Sense Ranking**: We studied if models are impacted by multi-sense concepts such as "period", and by their sense ranking (how frequent or rare those senses are). Surprisingly contextualized models, and specially CE models have no significant impact by this factor, suggesting that these models are very effective at deducing the correct sense based on their context. These charts also suggest that these models may be considering context even more than the words themselves. This is intuitive for Masked-Language-Models such as BERT, but not for others, such as GPT-2. Non-contextualized models are impacted by this factor, as expected.

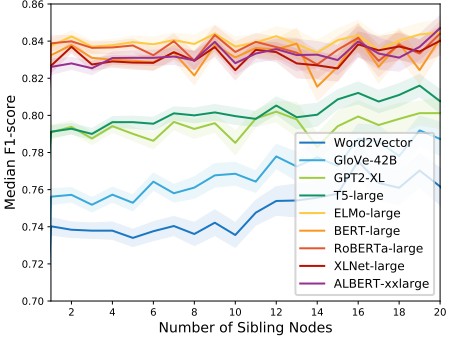
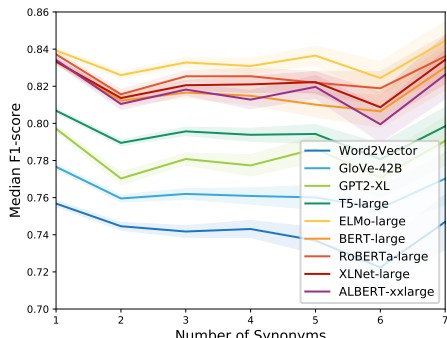

Figure 17: **Number of sibling nodes and number of synonyms**: We discover that these factors have very low impact on F1-scores. The fact that siblings are not confused leads to similar conclusions as the ones provided in Figure 16.

## D   ACCURACY COMPARISON

In Table 1 we analyzed F1 scores for each model and best layer. Here we provide an analogous table but reporting accuracies instead of F1 scores.

| Family | Model | Emb. Size All/Best Layer | Best Layer | Accuracy All Layers | Accuracy Best Layer |
|---|---|---|---|---|---|
| NCE | Word2Vec (Mikolov et al., 2013) | 300 / - | - | .7427 ± .0124 | - |
| | GloVe-42B (Pennington et al., 2014) | 300 / - | - | .7617 ± .0078 | - |
| GLM | GPT-2 (Radford et al., 2018) | 9984 / 768 | 6 | .7858 ± .0099 | .7756 ± .0097 |
| | T5-small (Raffel et al., 2019) | 7168 / 512 | 4 | .8075 ± .0078 | .8122 ± .0062 |
| | GPT2-xl (Radford et al., 2018) | 78400 / 1600 | 13 | .7910 ± .0102 | .7911 ± .0102 |
| | T5-large (Raffel et al., 2019) | 51200 / 1024 | 17 | .7988 ± .0113 | .8244 ± .0089 |
| CE | ELMo-small (Peters et al., 2018b) | 768 / 256 | 2 | .7872 ± .0106 | .7754 ± .0106 |
| | BERT-base (Devlin et al., 2019) | 9984 / 768 | 10 | .8234 ± .0100 | .8062 ± .0093 |
| | RoBERTa-base (Liu et al., 2019) | 9984 / 768 | 5 | .8349 ± .0083 | .8120 ± .0083 |
| | XLNet-base (Yang et al., 2019) | 9984 / 768 | 4 | .8267 ± .0092 | .8176 ± .0106 |
| | ALBERT-base (Lan et al., 2020) | 9984 / 768 | 12 | .8145 ± .0143 | .7970 ± .0089 |
| | ELMo-large (Peters et al., 2018b) | 3072 / 1024 | 2 | .8242 ± .0076 | .8245 ± .0075 |
| | BERT-large (Devlin et al., 2019) | 25600 / 1024 | 14 | .8221 ± .0115 | .8055 ± .0104 |
| | RoBERTa-large (Liu et al., 2019) | 25600 / 1024 | 13 | .8258 ± .0121 | .8189 ± .0067 |
| | XLNet-large (Yang et al., 2019) | 25600 / 1024 | 6 | .8215 ± .0105 | .8119 ± .0079 |
| | ALBERT-xxlarge (Lan et al., 2020) | 53248 / 4096 | 4 | .8215 ± .0086 | .8050 ± .0092 |

Table 5: Accuracies for every model in this study. This table is analogous to Table 1

## E   PRE-TRAINING CORPUS COMPARISON

| Family | Model | Pre-Training Corpus Size | |
|---|---|---|---|
| | | Tokens | Uncompressed Size |
| NCE | Word2Vec | 33B | 150GB* |
| | GloVe-42B | 42B | 175GB* |
| GLM | GPT-2 | 10B* | 40GB |
| | T5 | 180B* | 750GB |
| CE | ELMo | 0.8B | 4GB* |
| | BERT | 3.9B | 16GB |
| | RoBERTa | 38.7B* | 160GB |
| | XLNet | 32.9B | 140GB* |
| | ALBERT | 3.9B | 16GB |

Table 6: Pre-Training corpus sizes used for each one of the studied models. The official sources report corpus sizes in terms of number of tokens or uncompressed size in GB. The symbol * denotes values estimated by us based on official available information.

# F    ARE ALL THESE MODELS CORRELATED?

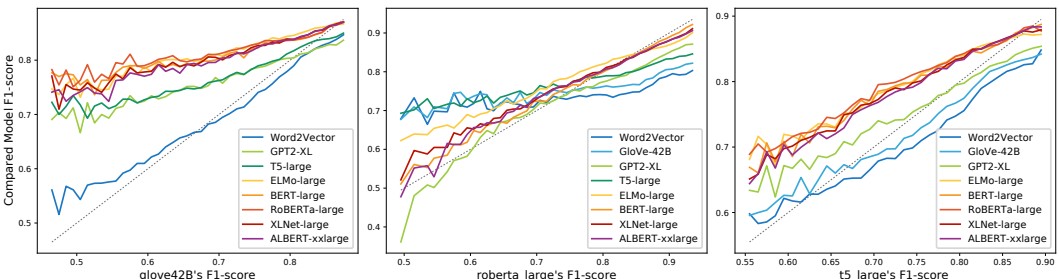

Figure 18: Analysis of F1-score correlations among models. A horizontal line in these charts would represent perfect independence.

|          | Word2Vec | GloVe | GPT-2 | T5    | ELMo  | BERT  | RoBERTa | XLNet | ALBERT |
|----------|----------|-------|-------|-------|-------|-------|---------|-------|--------|
| Word2Vec | -        | .6743 | .2138 | .4568 | .2453 | .1713 | .1582   | .2189 | .2276  |
| GloVe    | .6743    | -     | .3185 | .5071 | .2925 | .2541 | .2413   | .2896 | .3192  |
| GPT-2    | .2138    | .3185 | -     | .4675 | .6402 | .7351 | .7611   | .7212 | .7653  |
| T5       | .4568    | .5071 | .4675 | -     | .4826 | .3991 | .4122   | .4704 | .4597  |
| ELMo     | .2453    | .2925 | .6402 | .4826 | -     | .6146 | .6634   | .6674 | .6279  |
| BERT     | .1713    | .2541 | .7351 | .3991 | .6146 | -     | .8096   | .7655 | .7916  |
| RoBERTa  | .1582    | .2413 | .7611 | .4122 | .6634 | .8096 | -       | .7844 | .7890  |
| XLNet    | .2189    | .2896 | .7212 | .4704 | .6674 | .7655 | .7844   | -     | .7513  |
| ALBERT   | .2276    | .3192 | .7653 | .4597 | .6279 | .7916 | .7890   | .7513 | -      |

Table 7: Pearson correlation among concept-level F1-score obtained by different architectures. We only considered concepts with F1-scores between 0.55 an 0.9 to remove possible noisy outliers.

In this section, we study if concepts that are easy/hard to associate for one model are also easy/hard to associate for other models. We do this by analyzing the correlation between outputs using the semantic probing classifier. Charts in Figure 18 show the correlation among the models by plotting the F1-score of one model against the rest. We picked one model of each family: GloVe (Fig 18-left) RoBERTa-large (Fig 18-middle), and T5-large (Fig 18-right).

Figure 18-left reveals a strong semantic correlation between Word2Vec and GloVe, but weaker with the rest. Figure 18-middle shows a strong correlation between RoBERTa and other methods in its family, but weaker with models of the NCE family. Fig 18-right shows that T5 has the same difficulties as the rest of the methods, even showing some correlation with the NCE family of methods.

The charts in Figure 18 reveal that models are strongly correlated, specially within the same model family, suggesting that similar approaches lead to similar biases, regardless of their size. In other words, the concepts that were difficult in the past are, to great extent, still difficult for state-of-the-art models. Thus, novel inductive biases are needed to get closer to human semantic capabilities.

# G  ADDITIONAL F1-SCORES OF SEVERAL SEMANTIC CATEGORIES

| Category | W2V | GloVe | GPT-2 | T5 | ELMo | BERT | RoBERTa | XLNet | ALBERT |
|---|---|---|---|---|---|---|---|---|---|
| abstraction | .7142 ± .1277 | .7296 ± .1194 | .7224 ± .1897 | .7662 ± .0942 | .7808 ± .1203 | .7718 ± .1582 | .7759 ± .1537 | .7712 ± .1404 | .7635 ± .1732 |
| attribute | .7044 ± .1310 | .7213 ± .1237 | .7730 ± .0911 | .7649 ± .0863 | .8093 ± .0886 | .8094 ± .0998 | .8167 ± .0926 | .8064 ± .0891 | .8050 ± .0998 |
| communication | .6974 ± .1330 | .7251 ± .1224 | .7826 ± .0967 | .7587 ± .1083 | .8049 ± .0925 | .8246 ± .0979 | .8249 ± .0983 | .8066 ± .0987 | .8093 ± .1023 |
| group | .7068 ± .1320 | .6929 ± .1339 | .4972 ± .2955 | .7262 ± .1179 | .6821 ± .1711 | .6173 ± .2399 | .6256 ± .2305 | .6491 ± .2139 | .5858 ± .2745 |
| social group | .7497 ± .1058 | .7579 ± .1046 | .8155 ± .0883 | .7868 ± .0867 | .8312 ± .0707 | .8516 ± .0742 | .8552 ± .0698 | .8422 ± .0724 | .8556 ± .0819 |
| taxonomic group | .6920 ± .1306 | .6648 ± .1330 | .3030 ± .2025 | .6944 ± .1208 | .6011 ± .1583 | .4804 ± .2025 | .4921 ± .1903 | .5371 ± .1944 | .4277 ± .2305 |
| family | .7412 ± .1363 | .7213 ± .1244 | .3131 ± .2003 | .6691 ± .1276 | .5461 ± .1630 | .5626 ± .1537 | .5379 ± .1502 | .5733 ± .1626 | .5437 ± .1705 |
| genus | .6267 ± .0989 | .6040 ± .1127 | .2567 ± .1582 | .7156 ± .1001 | .6167 ± .1301 | .3696 ± .1857 | .4201 ± .1855 | .4555 ± .1853 | .2862 ± .1945 |
| psychological feature | .7256 ± .1122 | .7478 ± .1016 | .7829 ± .0904 | .7795 ± .0778 | .8163 ± .0851 | .8181 ± .0954 | .8229 ± .0930 | .8077 ± .0931 | .8208 ± .0915 |
| relation | .7264 ± .1304 | .7567 ± .1042 | .7612 ± .0809 | .7963 ± .0688 | .7679 ± .0878 | .7662 ± .0995 | .7649 ± .0982 | .7659 ± .0908 | .7727 ± .0929 |
| artifact | .7120 ± .1194 | .7389 ± .1068 | .7903 ± .0736 | .7868 ± .0676 | .8308 ± .0693 | .8249 ± .0742 | .8315 ± .0700 | .8319 ± .0702 | .8231 ± .0761 |
| covering | .7230 ± .1097 | .7510 ± .0970 | .7903 ± .0713 | .7878 ± .0606 | .8398 ± .0599 | .8392 ± .0706 | .8363 ± .0571 | .8393 ± .0576 | .8322 ± .0621 |
| instrumentality | .7064 ± .1219 | .7378 ± .1052 | .7930 ± .0728 | .7902 ± .0648 | .8308 ± .0676 | .8134 ± .0748 | .8337 ± .0691 | .8313 ± .0711 | .8233 ± .0763 |
| device | .7097 ± .1200 | .7435 ± .1009 | .7956 ± .0713 | .7899 ± .0667 | .8311 ± .0689 | .8198 ± .0701 | .8358 ± .0640 | .8326 ± .0675 | .8230 ± .0743 |
| causal agent | .7240 ± .1137 | .7398 ± .1101 | .7253 ± .1105 | .7751 ± .0757 | .8022 ± .0884 | .7453 ± .1093 | .7631 ± .1056 | .7788 ± .1035 | .7826 ± .0934 |
| person | .7208 ± .1135 | .7351 ± .1113 | .7207 ± .1132 | .7735 ± .0746 | .8022 ± .0854 | .7379 ± .1101 | .7585 ± .1054 | .7779 ± .1053 | .7826 ± .0937 |
| living thing | .7295 ± .1112 | .7421 ± .1078 | .7300 ± .1004 | .7862 ± .0749 | .8187 ± .0850 | .7593 ± .0997 | .7823 ± .0922 | .7907 ± .0928 | .7758 ± .0909 |
| animal | .7349 ± .1049 | .7391 ± .1028 | .7515 ± .0829 | .7837 ± .0714 | .8389 ± .0785 | .7914 ± .0801 | .8179 ± .0701 | .8135 ± .0737 | .7781 ± .0857 |
| plant | .7445 ± .1044 | .7608 ± .0989 | .7288 ± .0842 | .8168 ± .0653 | .8404 ± .0692 | .7679 ± .0830 | .7962 ± .0688 | .7986 ± .0746 | .7645 ± .0861 |
| matter | .7402 ± .1117 | .7633 ± .1009 | .7582 ± .0834 | .8002 ± .0662 | .7756 ± .0886 | .7645 ± .0959 | .7614 ± .0941 | .7636 ± .0908 | .7685 ± .0938 |
| part | .7532 ± .1000 | .7759 ± .0852 | .7540 ± .0772 | .8051 ± .0595 | .7580 ± .0844 | .7499 ± .0977 | .7441 ± .0947 | .7526 ± .0880 | .7610 ± .0907 |
| substance | .7560 ± .0967 | .7791 ± .0809 | .7508 ± .0755 | .8073 ± .0567 | .7542 ± .0828 | .7436 ± .0952 | .7370 ± .0919 | .7477 ± .0862 | .7580 ± .0889 |

Table 8: F1-score and standard deviation of several semantic categories evaluated on the larger version of each model. Each value was computed by considering the F1-score of all the concepts that belong to the analyzed category. This is not an extensive list and categories are somewhat imbalanced. Categories were selected based on the number of sub-categories they contained.

