# OpenReview forum: "Tracking the progress of Language Models by extracting their underlying Knowledge Graphs"
_ICLR.cc/2021/Conference — Reject_

### Official Review · AnonReviewer1 · 2020-10-25
**Good analysis, but limited contribution**

**Rating:** 4
**Confidence:** 2

**Review:**

This paper analyzed how well the previously proposed pre-training models could encode the underlying knowledge graph by defining probing classifiers. The probe classified is trained on top of the pre-trained contextual presentation models, such as non-contextualized word embeddings (e.g., Glove), contextualized word embeddings (e.g., BERT), and generative language models (e.g., GPT-2), and tried to reconstruct the structure of the knowledge graphs.

This paper is well-written. Readers will easily understand what this paper did and tried to reveal. While it was not sufficiently clear why this paper adopts this approach, I think the idea of the probing classifiers and knowledge graph construction is a reasonably good idea to reveal how well the pre-training models encode the underlying knowledge graph.

The cons of this paper are the lack of a profound analysis of the experimental results. In Section 5, this paper tried to reveal what knowledge the existing pre-trained models work well or not well by using some statistics (e.g., the relative depth). While I think this approach is also good, readers will need more detailed information about analysis results to inspire new ideas to improve semantic learning abilities. For example, the number of samples in each concept depth and wordnet distance between concepts changes. Therefore, it is better to estimate confidence intervals for readers to precisely understand how the differences of the median F1-scores is important (or not important) in each depth or distance. Second, it is better to analyze which semantic category the pre-trained methods work well or not well. I guess that the concept depths and frequencies hugely change depending on the concept categories. It is helpful if this paper also elucidates which semantic cateogy the existing method work well and not well. Thirdly, it is better if this paper shed light on how readers can improve the semantic learning abilities based on these results. Without these proposals, I think the contribution of this paper is limited.

---

> ### Author Response · Authors · 2020-11-21
> **Extensive modifications. All suggestions addressed**
>
> The paper had extensive modifications to address reviewers suggestions, resulting in improved analysis and increased clarity. In particular, regarding your comments:
>
> * We have conducted a rigorous analysis of our paper, reorganizing information and highlighting the most consistent findings in order to clarify the message of the work. In particular, we include a new Table that summarizes our main findings and includes links to the corresponding supporting evidence. We believe that this new version of the paper is more clear than the previous version. Thanks for the suggestions. They allowed us to improve our paper.
> * Section 5 was modified to provide a deeper analysis to inspire new ideas on how to improve semantic abilities, following all reviewer suggestions. We now pay attention to the most consistent findings of the paper, providing actionable insights for researchers and practitioners who may want to improve the semantic abilities of their models. We also added confidence intervals to Figure 4. Thanks for pointing it out.
> * We added Section 7, which includes further analysis and discussion on the implications of these findings, shedding light on how these findings can be used to improve semantic abilities. We also highlight potential applications of our findings.
> * As suggested, we added to section 5 a new analysis at a category-level. Specifically, we include a new table that incorporates results for several semantic categories, indicating the performance of the different models. This analysis clarified the message of the section leading valuable insights. We also expand this analysis by including information in Appendix G.

---

### Official Review · AnonReviewer3 · 2020-10-28
**Interesting technique, needs stronger message and more rigor**

**Rating:** 5
**Confidence:** 4

**Review:**

The paper 1) introduces a method to use three types of text embedding methods (non-contextual, contextual, LM -based) to predict  word relatedness (as a binary classification problem) for pairs of words in wordnet 2) uses these relatedness scores to build proxies of the Wordnet graph 3) carry out experiments based on the two bullet points to compare semantic understanding abilities of the aforementioned embedding methods.

Major comments:
* The paper carries out quite a few experiments with weakly connected goals. It looks like a combination of miscellaneous results based on a common (and limited) technique, rather than delivering a coherent message with interrelated takeaways from follow-up experiments.
* The technique to probe models is quite restricted in that it is centered around single word concepts. Given that contextualized models are utilized, it seems like a rather handicapped investigation of very powerful models.
* In section 4.3, authors try to make a point about correlations on visuals. This is a dangerous approach, and it would be much better to rely on numerical summaries of correlations. In fact, it is extremely hard to judge correlation by looking at pictures, because correlation needs to take into account the variability in an F1 metric with the other axis kept constant (only means are shown). A curve with less slope on Figure 3 might indicate a much higher degree of co-movement with the metric on the X axis if the randomness in y axis wrt at any point in x axis is very low. Authors should revisit statistical correlation, and preferably revamp this section. That said, I'm not quite convinced that it is a publication-worthy result to say similar methods (and each of the 3 buckets is very similar within) produce similar concept relatedness scores.
* It's hard to understand how the proposed probing classifier is different than concatenating $M(x)$ and $M(y)$ and directly applying MLP on it. One can choose an MLP with custom first hidden layer size and activation function that would be functionally equivalent to what's being proposed.
* There is definitely truth to the title, but I'd suggest not conflating the term "knowledge graph", which traditionally represents actual world knowledge, and not lexical databases.

Minor comment
* Please write something under section 3 (before 3.1). Given that it's not clear on a first pass that you're introducing two different methods in 3.2 and 3.3, the empty space is a good opportunity to tell the reader about this fact.

---

> ### Author Response · Authors · 2020-11-21
> **Extensive modifications. Suggestions addressed**
>
> * The probe is not restricted to non-contextualized single word concepts. We guess that we did not explain this point clearly enough in the previous version, thus we updated the document and Figure 1 to make it more clear. Our experiments take advantage of a dataset that contains annotations of the appearances of WordNet concepts in full sentences. Using this dataset, we can obtain the embedding of a target concept using a context-aware model. We just run the model over a sentence mentioning that particular concept. Then, we keep only the embedding that correspond to the token of the mentioned concept (or the first of them in the case of concepts longer than one token).
> * We added an introduction below Section 3 (before 3.1) as suggested, along with suggestions from Reviewer4.
> * We applied a linear transformation to M(x) and M(y) before the MLP mainly to standardize the dimensionality across the different models. We clarified this point in the paper. As you pointed out, we agree that concatenating M(x) and M(y) and applying a custom sized MLP is also a viable way to implement the probing classifier. We believe that it would lead to similar results.
> * As suggested, Section 4.3 was removed and replaced by a quantitative analysis of models performance across semantic categories (please see Section 5 in the new version of the paper), leading to valuable insights that integrates well with the message of the rest of the paper.
> * You mentioned that the paper seemed to have weakly connected goals. The paper is now organized around three main topics: 1) Ability of the models to encode the semantic information in Wordnet (Section 4); 2) Analysis of the model strengths and weaknesses to encode this knowledge (Section 5); and 3) Location where this knowledge is primarily encoded inside each model architecture (Section 6). We updated the whole paper and discussions to convey a more coherent message. In addition, we included a new section (please see Section 7) where we highlight the main findings of the paper. We also suggest how to transform these findings in actionable insights for future work.

---

### Official Review · AnonReviewer2 · 2020-10-28
**The authors' contribution is an extensive study on how different language models incorporate semantic knowledge on the concept level.**

**Rating:** 6
**Confidence:** 4

**Review:**

 The authors conduct a study investigating how different language models incorporate semantic information in their respective learned representations. Investigating language models on their performance in concept-level tasks is motivated by the importance of the ability to organize and understand concepts in human intelligence. Another motivation is that other studies on the semantics in language models are not conclusive according to the authors, especially in determining where the semantic knowledge lies within the language models.
The study is conducted by using a semantic probing classifier which – in short – is trained to determine whether two words (inputted as learned representations from the language models) are semantically related (according to WordNet) or not. This classifier also aids in recreating a sampled knowledge sub-graph from WordNet.
The experimental section contains the evaluation of the two tasks, firstly the classification as described above and secondly the KG reconstruction.
The main findings of the study can be summarized as follows:
- The authors show experimentally that models coming from the same family are strongly correlated
- The authors show the experimental outcomes of the tasks mentioned above
- The authors show how the individual layers of the language models contribute to the underlying knowledge
- The authors show for all models how they are affected by different semantic factors (9 different factors, including number of senses, graph depth etc.)
The paper is clearly written and understandable and includes enough details to understand the implementation of the semantic probing classifier. The appendix contains detailed outcomes of the different experiments which, together with the result section, give a good overview of the experimental results.

My recommendation is towards acceptance of the paper, because the authors contribute to a more detailed understanding on how language models incorporate semantic knowledge and where this knowledge might be located within the models. Exploiting those findings could potentially lead to an improvement on future models. Also, the findings per se give more insight on how the internals of large models process information, which is a step towards a more explainable AI.

I have one question regarding the inter-comparability of the models; were the tested language models all trained on the same unlabeled textual data (or on data of comparable size), or did you use pre-trained models that were published alongside their respective papers?

---

> ### Author Response · Authors · 2020-11-21
> **Please more feedback**
>
> * We used pre-trained models provided by the original papers. We updated the paper to make this fact clearer. We also enhanced discussion and analysis in section 7 and appendix E regarding the low impact of pre-training corpus sizes.
> * May we ask why you changed your Rating from 7 to 6? We would greatly appreciate if you could point out the elements that you consider that require more work. The suggestions of the other reviewers were already addressed, resulting in improved analysis and clarity in the message of the paper.

---

### Official Review · AnonReviewer4 · 2020-10-29
**ICLR - TRACKING THE PROGRESS OF LANGUAGE MODELS BY EXTRACTING THEIR UNDERLYING KNOWLEDGE GRAPHS**

**Rating:** 6
**Confidence:** 4

**Review:**

Summary

This work addresses the question about how pre-trained language models encode semantic information. It adapts the methodology proposed in Hewitt & Manning (2019) for syntax to semantics, using the WordNet structure instead of a syntactic structure of a sentence to encode distances among word representations. The paper analyzes how embedding models encode suitable information to recreate the structure of WordNet. The study also shows evidence about the limitations of current pre-trained language models, demonstrating that all of them have difficulties to encode specific concepts.

Quality

The proposed idea is very interesting, but the paper does not give a complete picture of what probing tasks can show and what their limitations are. The contribution of the paper is not clear. What can we learn from the experiment of the paper? How can we improve current language models? How can we exploit the distilled information?

Missing reference for semantic probing tasks

Yaghoobzadeh, Yadollah, et al. "Probing for Semantic Classes: Diagnosing the Meaning Content of Word Embeddings." Proceedings of the 57th Annual Meeting of the Association for Computational Linguistics. 2019.
Peters, Matthew, et al. "Dissecting Contextual Word Embeddings: Architecture and Representation." Proceedings of the 2018 Conference on Empirical Methods in Natural Language Processing. 2018.

Missing references for the usefulness of probing tasks

Saphra, Naomi, and Adam Lopez. "Understanding Learning Dynamics Of Language Models with SVCCA." Proceedings of the 2019 Conference of the North American Chapter of the Association for Computational Linguistics: Human Language Technologies, Volume 1 (Long and Short Papers). 2019.

A recent one paper on the usefulness of probing tasks

Ravichander, Abhilasha, Yonatan Belinkov, and Eduard Hovy. "Probing the Probing Paradigm: Does Probing Accuracy Entail Task Relevance?." arXiv preprint arXiv:2005.00719 (2020).

Clarity

-       The paper is clear and well written even if some references about the soundness of probing tasks are missing (see above) and a related discussion is missing too. In fact, the results of probing tasks have been questioned (see the references above) because it is not clear if the use of supervision allows the representation to adapt to the task.
-       The related work section is a list of contributions and successes in probing tasks without a clear narrative.
-       WordNet is not introduced
-       Many acronyms are not defined (e.g.: WSD, MLP)
-       I think that this part is important and should be clarified (last paragraph of Section 3.2): “Tests based on linear transformations such as that proposed by Hewitt & Manning (2019) did not allow us to recover the WordNet structure, which indicates that the subspaces in which the word embeddings models encode the semantics are not linear”. Intuitions or even hypotheses about this behaviour are not given.

Originality

-       The analysis includes recent models such as ALBERT and T5.
-       The idea of using the WordNet taxonomy to adapt the model proposed in Hewitt & Manning (2019) is very interesting

Significance

The proposed idea is very interesting and also the methodology is sound, but the conclusions are weak:
- It is intuitive that it is more difficult to encode distant relations than others.
- The fact that models in the same family have similar results is not discussed
- It is not explained why only the Princeton WordNet Gloss Corpus has been used and not larger datasets annotated with WordNet senses such as SemCor.
- Usually the models are evaluated at each layer, here all the layers are concatenated making it more difficult to understand where semantic information is stored.

---

> ### Author Response · Authors · 2020-11-21
> **All suggestions addressed**
>
> *  We included the suggested references on probe methods. We also improved the discussion on the soundness and limitations of probe methods that arise from these new references (please see the first paragraphs in Section 3 of the new version of the paper).
> * We fixed narrative and included missing definitions as suggested (e.g., Wordnet).
> * We included hypotheses about why semantics are not encoded in linear subspaces, but require a non-linear model to be extracted (please see the last paragraph of section 3.2).
> * We included a new Section (please see Section 7) with further discussion of the results and implications of the findings, to shed light on how these findings can be used to improve semantic abilities of future works.
> * We used the Princeton WordNet Gloss Corpus because it covers around 34000 different noun synsets with more than 42000 different lemmas. SemCor can also be useful for this task, but it is a little less diverse covering around 13000 noun synsets and 26000 lemmas.
> * Layer-level results are now in sections 4.1 and 6, leading to new insights. Thanks for the suggestion.
> * You correctly pointed out that it is not surprising that distant relations are more difficult to encode than others. To improve the relevance of the information included in the paper, we replaced the corresponding sub-charts of Figure 3 with a different set of charts that illustrate less-intuitive but  relevant information to support the discussion. Specifically, we include: F1 v/s "Number of Child Nodes", "Number of Senses", "Sense Ranking" and "Number of sibling nodes". The charts about distances is now included in Appendix C.
> * We included a new discussion throughout sections 5, 6 and 7 regarding that models in the same family have similar results, as suggested.

---

### Decision · Program_Chairs · 2021-01-07
**Final Decision**

**Decision:**

Reject

**Comment:**

This work addresses the problem of understanding how pre-trained language models are encoding semantic information, such as WordNet structure. This is evaluated by recreating the structure of WordNet from embeddings. The study also shows evidence about the limitations of current pre-trained language models, demonstrating that all of them have difficulties to encode specific concepts.

pros:
- good idea to reveal how well the pre-training models encode the underlying knowledge graph
- detailed understanding on how language models incorporate semantic knowledge and where this knowledge might be located within the models
- experiments show that models coming from the same family are strongly correlated
- the paper shows how individual layers of the language models contribute to the underlying knowledge
- analysis of the different semantic factors (9 different factors, including number of senses, graph depth etc.)
- paper is clearly written and understandable and includes enough details to understand the implementation of the semantic probing classifier.

cons:
- weakly connected goals, response from reviewers is string around 3 main topics, which is seen as many for a single scientific paper. It would be easier to focus only on one topic and make a clear conclusion,
- single word concepts while CE models are powerful in context,
- lack of a profound analysis of the experimental results
    - hard to understand which semantic category the pre-trained methods work well or not well,
    - clarification about the improvement of the semantic learning abilities based on these results.

Several of the identified issues have been answered in the author's rebuttal, however, the paper would still need more work to be accepted. Note also that the bar a this year ICLR conference is high and we encourage the authors to submit their updated work again at the next conference.